Joint estimation of crown of thorns (Acanthaster planci) densities on the Great Barrier Reef

MacNeil M. Aaron a.macneil@aims.gov.au 1 2 3
Mellin Camille 1 4
Pratchett Morgan S. 2
Hoey Jessica 5
Anthony Kenneth R.N. 1
Cheal Alistair J. 1
Miller Ian 1
Sweatman Hugh 1
Cowan Zara L. 2
Taylor Sascha 6
Moon Steven 7
Fonnesbeck Chris J. 8
1 Australian Institute of Marine Science , Townsville , Queensland , Australia
2 ARC Centre of Excellence for Coral Reef Studies, James Cook University , Townsville , Queensland , Australia
3 Department of Mathematics and Statistics, Dalhousie University , Halifax , Nova Scotia , Canada
4 The Environment Institute and School of Biological Sciences, University of Adelaide , Adelaide , South Australia , Australia
5 Great Barrier Reef Marine Park Authority , Townsville , Queensland , Australia
6 Queensland Parks and Wildlife Service (Marine Parks), Queensland Deptartment of National Parks, Sport and Racing , Brisbane , Queensland , Australia
7 Association of Marine Park Tourism Operators , Cairns , Australia
8 Department of Biostatistics, Vanderbilt University Medical Center , Nashville , TN , United States of America
Reimer James
Electronic publication date: 2016 Aug 31
Publication date: 2016
Volume: 4
Electronic Location ID: e2310
Received 2016 May 18; Accepted 2016 Jul 11
Copyright: ©2016 MacNeil et al.
Copyright year: 2016
Copyright holder: MacNeil et al.
License: This is an open access article distributed under the terms of the Creative Commons Attribution License, which permits unrestricted use, distribution, reproduction and adaptation in any medium and for any purpose provided that it is properly attributed. For attribution, the original author(s), title, publication source (PeerJ) and either DOI or URL of the article must be cited.
License URL: https://creativecommons.org/licenses/by/4.0/

Keywords: CoTS, Bayesian analysis, Monitoring, Mark-recapture, Data integration

Funding: Australian Institute of Marine Science Queensland Government This work was supported by funding from the Australian Institute of Marine Science and from the Queensland Government via an Accelerate Partnership under the Department of Science, Industry, Technology and Innovation. The funders had no role in study design, data collection and analysis, decision to publish, or preparation of the manuscript.

==============================
Crown-of-thorns starfish (CoTS; Acanthaster spp.) are an outbreaking pest among many Indo-Pacific coral reefs that cause substantial ecological and economic damage. Despite ongoing CoTS research, there remain critical gaps in observing CoTS populations and accurately estimating their numbers, greatly limiting understanding of the causes and sources of CoTS outbreaks. Here we address two of these gaps by (1) estimating the detectability of adult CoTS on typical underwater visual count (UVC) surveys using covariates and (2) inter-calibrating multiple data sources to estimate CoTS densities within the Cairns sector of the Great Barrier Reef (GBR). We find that, on average, CoTS detectability is high at 0.82 [0.77, 0.87] (median highest posterior density (HPD) and [95% uncertainty intervals]), with CoTS disc width having the greatest influence on detection. Integrating this information with coincident surveys from alternative sampling programs, we estimate CoTS densities in the Cairns sector of the GBR averaged 44 [41, 48] adults per hectare in 2014.

Introduction

Outbreaking pests generate extensive environmental degradation and billions of dollars in ecosystem damage annually for resources such as crops (Oerke, 2006), forests (Aukema et al., 2011), and fisheries (Knowler & Barbier, 2000). Among the most ecologically and economically costly are outbreaks of native crown-of-thorns starfish (CoTS; Acanthaster spp; Carter, Vanclay & Hundloe, 1988), a common coral-eating starfish that outbreaks on many Indo-Pacific coral reefs (Pratchett et al., 2014). CoTS are considered a pest due to their destructive impacts on coral populations when outbreaks occur. The causes of CoTS outbreaks remain largely unknown, although prominent theories include decreased predation due to fishing of key species (Sweatman et al., 2008) and increased larval survivorship due to nutrient pollution (Fabricius, Okaji & De’ath, 2010). Of all the major disturbances to coral reefs (e.g., storms, coral bleaching, fishing, pollution, nutrients, and disease) only CoTS outbreaks have the potential to be actively locally managed without negatively impacting other resource users. Yet there is insufficient information available about many aspects of CoTS population dynamics and life history (Pratchett et al., 2014) to accurately predict the success of management intervention.

Of the many factors needed to fully understand CoTS population dynamics—including fecundity, larval survival, and settlement—basic information on adult densities is lacking due to inconsistencies in sampling methods that can underestimate local abundance. A key gap in the accuracy of survey data is detectability—the probability of observing individual starfish underwater, given their presence in a surveyed area. Without some estimate of detectability, survey data cannot be indexed to true abundance, limiting the inferences that can be made concerning CoTS population dynamics. Detectability estimation has been widely used among terrestrial and freshwater wildlife studies, including for birds (MacKenzie et al., 2002), mammals (Gu & Swihart, 2004), and fish (Peterson & Dunham, 2003), but these methods have not been widely applied to coral reefs (but see Kulbicki & Sarramégna, 1999; MacNeil et al., 2008). Quantification of adult CoTS detectability is important for population modeling, spread dynamics, and assessing the feasibility or efficacy of control actions.

K-sample mark-recapture methods for estimating detection probabilities are well suited to repeat visual sampling of most site-attached reef species. These include the application of one or more identifiable marks to individuals within a specific area over a series of K capture occasions, with each observed presence or absence recorded in a K dimensional array of ones and zeros that make up an individual’s capture history (i.e., 0, 1, 1, 0, 1, 1 for K = 6). In aggregate, an observed population of capture histories allows a distribution of detection probabilities to be constructed and used to estimate how many individuals remained unobserved during sampling.

The Great Barrier Reef (GBR) consists of around 3,000 interconnected reefs stretching more than 2,300 km along the coast of northeastern Australia. There is strong national and international pressure to protect the GBR from increasing degradation due to a series of compounding human and natural disturbances that have eroded average reef conditions over the past 25 years (De’ath et al., 2012). While some disturbances are linked to climate change (e.g., storms, bleaching; Osborne et al., 2011) and water quality (Fabricius, Okaji & De’ath, 2010), CoTS remain a major source of coral mortality (De’ath et al., 2012). As such, there is considerable interest in accurately estimating reef-wide CoTS abundance to (a) assess the feasibility of reef-wide controls and (b) project CoTS impacts in terms of ongoing reef degradation.

A key problem in understanding broad-scale CoTS population dynamics on the GBR is that agencies use different survey methods to record the presence, absence, or density of CoTS using different methods that may not be directly comparable. For example, the Queensland Parks and Wildlife Service (QPWS) and Great Barrier Reef Marine Park Authority’s (GBRMPA) joint Field Management Program (FMP) and the Association of Marine Park Tourism Operators (AMPTO) CoTS Control Team conducts reef health surveys to record CoTS abundance, their feeding scars, and the health of their corals prey. This program acts as an early warning system and surveillance tool to help inform cull efforts. AMPTO, in partnership with managing agencies monitors coral cover, culls CoTS and records the number killed within specific areas, with approximately $2.5 million (Australian dollars) committed to the protection of coral in high value tourism areas in 2014–15 (ReefPlan, 2014). The Australian Institute of Marine Science (AIMS) long term monitoring program (LTMP) uses a third method of manta-tow surveys to detect whether a reef is suffering CoTS outbreaks above natural densities (Sweatman et al., 2008). Currently these different methods of sampling CoTS, while similar in some respects (e.g., AMPTO, AIMS, and the FMP program all use manta tow), cannot be reconciled. Without knowing the true density of CoTS within a survey area the potential bias in each method remains unknown.

To improve abundance estimation for CoTS on the GBR we utilize a K-sample mark-recapture model to estimate detection and infer the density of CoTS in a northern section of the GBR, using covariates thought to impact CoTS detection underwater and a joint model for integration of FMP and AMPTO datasets.

Methods

Data and analysis consisted of two components. The first involved a dedicated mark-recapture study, conducted by three key data-providing organizations (AIMS, FMP, and AMPTO) that enabled us to estimate the detectability of adult CoTS and the underlying true adult (>14 cm) density within a set of study sites (hereafter ‘model-calibrated density’). We then used these values to calibrate broad-scale count (FMP) and kill (AMPTO) datasets and estimate the density of CoTS among reefs in the Cairns Sector of the GBR in 2014. All observations were made under Great Barrier Reef Marine Park Authority Permit G06/19994.1.

Mark-recapture surveys

A dedicated mark-recapture survey was conducted by AIMS, FMP, and AMPTO divers in May 2014 at eight 50 × 5 m transects (sites) spread between two reefs–Undine (Reef 16-023, n = 3) and Rudder (Reef 16-020a, n = 5)–in the Cairns Sector of the GBR (Fig. 1A); note that reef identification codes are unique identifiers for reefs within the GBR. Sites were selected opportunistically, based on known CoTS outbreaks and were accessible by survey boats. All fieldwork was conducted under the Australian Institute of Marine Science research permit issued by the Great Barrier Reef Marine Park Authority. Surveys were conducted on days with similar sea-state conditions, at depths where sea state was assumed not to have affected detectability.

Figure 1 Sampling scheme for crown-of-thorns starfish (CoTS) mark-recapture study.

(A) Study reefs on the Great Barrier Reef; (B) multiple tagged CoTS #62 from the study; (C) parameter-expanded data augmentation (PXDA) matrix of observed (1 to n), unobserved (n + 1 to N), and not present (N + 1 to M) individuals within the study superpopulation (M); (D) reef schematic showing partitioning of reef (r) perimeter into manta-tow sections (s), delineated by radiating straight lines. yi denotes the number of capture occasions over which an individual was observed; zi indicates an individual was observed in any capture occasion.

Within each site, FMP divers conducted initial surveys of the reef surface at between seven to nine meters depth, using standard FMP 5 m-radius point count methods (“Reef Health and Impact Surveys (RHIS)”; Beeden et al., 2014) to record the local abundance of CoTS within a total area of 400 m2 (n = 5). These abundances were converted to densities (CoTS/m2) for comparison with estimates of true densities estimated by mark-recapture.

Next, AIMS divers, using effort and swim speeds typical of AIMS Long-Term Monitoring Program methodology (LTMP), scanned each site looking for CoTS. When a new starfish was spotted it was tagged with two or three clothing tags (plastic t-tags) that included a unique identification number (Fig. 1B). Each transect was re-surveyed K = 6 times during both day (60% of observations) and night, by one of three AIMS observer teams. As tag shedding has been reported to be a problem by other researchers we confined sampling of each transect to a single 24 h period, over which CoTS were assumed not to lose tags. In addition, the total percentage of hard coral cover within the transect was recorded as a transect-scale covariate thought to affect detectability.

Finally, after FMP and AIMS surveys were completed, AMPTO divers visited each site and proceeded to kill all observed CoTS within the survey area, using standard cull procedures and recording the number of CoTS killed per unit time of effort (CPUE).

Mark-recapture submodel

To quantify CoTS detectability and estimate their model-calibrated density within the survey areas we employed a parameter-expanded data-augmentation (PXDA) approach for closed-population mark-recapture (Royle & Dorazio, 2012). This flexible class of models allows for individual, spatial, and time-varying heterogeneity in the detection probability of CoTS that is directly integrated into estimates of population size at a given site (Nj). The basic strategy is to use information from the individuals observed at a given site (nj) to estimate the number of unobserved individuals (uj = Nj − nj). By conducting repeated sampling of a fixed area and tagging individual CoTS, a set of sampling histories can be built up that records if individual i was observed (recorded as 1) or not (recorded as 0), across a set of k capture occasions (i.e., yij = 1, 0, 0, 1, 1, 0; Fig. 1C). The resulting matrix of observed individuals allows an average probability of detection to be estimated (ϕ), given covariate information within a linear model.

PXDA works by augmenting the observed set of individual sampling histories with an arbitrary number of unobserved sampling histories to create a superpopulation of “pseudo-individuals” (M; Fig. 1C). This superpopulation comprises three components: the observed individuals; a group of unobserved, but present, individuals; and a group of individuals not present in the sample area. The PDXA approach uses probability to assign each of the augmented individuals into unobserved or not present groups using the characteristics from the observed part of the population. Note that this approach will underestimate population size if the detectability of the true unobserved population is different from the observed population due to, for example, physical or behavioural differences.

Unbiased estimation of CoTS densities on a given reef likely depends on habitat characteristics in the survey area and the ability of divers to detect, and therefore count, CoTS underwater. Potentially important habitat factors include substrate type, structural complexity, water clarity, depth, and time of day. Body size and level of activity also likely influence the detection of individual CoTS, especially given that large adult starfish are generally less cryptic than juvenile individuals.

To account for the influence of habitat factors on detection we developed a hierarchical model using available covariates for detection including site-scale hard coral cover (HC), observer team (OT), the disc-width (DW) of individual CoTS, presence of a tag (PT), and a dummy variable for surveys conducted at night (NI), with corresponding individual, sampling-occasion, and transect-scale parameter estimates (γ1, a1,2,3,4).

The disc-width individual covariate adds a slightly complicating step to the PXDA approach in that the disc-widths of some observed CoTS were missing and for unobserved individuals are unknown. However, our Bayesian approach allowed us to integrate over this uncertainty by estimating the missing lengths from the observed lengths within the model. The full model was therefore: (1) yijk∼BernZiϕijk

(2) Zi∼Bernψ

(3) ϕijk=invlogita0j+a1DWA,j+a2NIjk+a3PTjk+a4OIjk

(4) a0j∼Nμj,τ0

(5) μj=γ0+γ1HC

(6) DWA,j=observed,DWiunobserved,NμDW,τ1IDWi>0

(7) ψ∼U0,1

(8) μDW∼U1,300

(9) γ0,γ1,a0..3∼N0.0,0.001

(10) τ0,τ1=σ0−2,σ1−2

(11) σ0,σ1∼U0,1000.

This model includes a ‘factor potential’ I(DWi > 0), an arbitrary indicator function that constrains unobserved values of DW to be positive (Lauritzen et al., 1990). Note that distributions within the model notation above are specified by their precision. Within the model, the detection component influences the probability of observing a zero (1 − ϕijk) among the augmented population and the total estimated population (N) is given by the posterior ∑Zi. Similarly the total estimated population for each site Nj is given by ∑Zij, and the corresponding density per hectare as ρj = 40(∑Zjk).

Data-calibration submodel

With the true (known) population (Nj) and density (ρj) at each site estimated from the mark-recapture submodel, we developed an additional submodel to calibrate the FMP counts and AMPTO CPUE data observed at each location, in slightly different ways.

FMP calibration

First, for the FMP data we assumed, due to repeated evidence of comparability between transects and point counts (Samoilys & Carlos, 2000), that detectability would be the most important source of potential bias relative to the known CoTS population size. Therefore, bias BFMP was estimated by dividing densities observed by the FMP team at each site by the average site-level detectability (ϕj ¯), checking for bias relative to the model-calibrated density: (12) BFMP=OBSFMPϕj ¯−ρj

with BFMP being centered on zero taken as substantial evidence that detectability accounts for potential bias present in the FMP observations.

AMPTO calibration

While FMP data is broadly comparable to the mark-recapture data in form, the AMPTO CPUE data is a different measure of abundance, where the total number of CoTS observed is standardized over some unit of effort; in this case, the duration of the dive over which the counts were made. CPUE is known from fisheries science to be a notoriously inconsistent index of abundance (Hartley & Myers, 2001) that can either decline quickly (hyperdepletion) or remain high (hyperstability) as true abundance declines. Hyperstability is expected to occur where populations are highly clustered and handling time to catch or kill individuals dominates the time spent searching for them. A common model for CPUE data is: (13) CPUEt=qNtB

where CPUE is proportional to true abundance (Nt) given a catchability coefficient (q) and scaling parameter (B). In general, both q and Nt are difficult to estimate; however, given the mark-recapture estimates from our finite survey area, catchability is given by detectability and true abundance is assumed known. Therefore, we estimated the relationship between log(CPUE) and true abundance as: (14) logCPUEjk∼Nθj,τθ

(15) θjk=logϕj ¯+B∗logNj

(16) B∼U0.0,10

(17) τθ=σθ−2

(18) σθ∼U0,1000.

This parameterization essentially defines a log-Normal relationship between true densities and CPUE, estimating parameters on a log–log scale.

Regional surveys

As part of an Australian government-funded effort to control CoTS on high-value tourism and ecological reefs and support the Queensland tourism industry, AMPTO devotes considerable effort to protecting key tourism reefs by applying a lethal injection to the starfish (Rivera-Posada et al., 2011). In 2014, this program of targeted control was expanded beyond the high-priority tourism reefs in the Cairns sector to include a subset of ‘super-spreader reefs’ identified as being highly connected by water currents that receive and then spread CoTS larvae more widely than other reefs (Hock et al., 2015).

Operational decisions about where AMPTO can direct their control efforts beyond the primary tourism reefs comes from GBRMPA and the Reef and Rainforest Research Centre, in consultation with the AMPTO Project Manager. GBRMPA monitor average coral cover over time in combination with CPUE to determine if thresholds are breached (e.g., coral cover declining and/or CPUE increasing) and when reefs require re-visitation. In addition, surveys of COTS and coral health from the FMP provide more source information on the distribution of CoTS within the Cairns sector than among other sectors. Therefore, across much of the northern GBR there are multiple interconnected surveys in which either CoTS counts or CoTS CPUE information are collected; both are informative about the density of CoTS on the GBR. A key step in integrating these various datasets was to use the mark-recapture study information to calibrate these data sources and use the calibration to jointly estimate CoTS densities.

Joint zero-inflated survey model

Because the FMP conducts a two-part monitoring program whereby entire reefs are coarsely surveyed to detect the presence of outbreaks using manta-tows and counts of CoTS outbreaks are made by smaller-scale underwater visual count (UVC) surveys, we developed a two-part mixture model that included explicit outbreak (occupancy) and count (abundance) components for each manta-tow section surveyed on each reef (Fig. 1D). We parameterized this mixture using a zero-inflated Poisson model (ZIP): (19) yrs∕ϕ ¯∼0withprobabilityπrsPoisλrswithprobability1−πrs

with the response being the observed count yrs on each manta-tow segment (s) within each reef (r), calibrated by the average detectability (ϕ ¯) estimated from the mark-recapture model.

The first submodel component was a hierarchical zeros model to estimate probability of a CoTS outbreak—defined as three or more CoTS or feeding scars per manta-tow (Doherty et al., 2015)—occurring on any given segment: (20) logitπrs∼Nβr,τβr

(21) βr∼Nβ0+β1DLI,τβ0

(22) β0,1∼N0.0,0.001

(23) τβ,τβr=σβ−2,σβr−2

(24) σβ−2,σβr−2∼U0,1000.

The model included the distance of each reef from Lizard Island (DLI), reflecting the hypothesized source of CoTS outbreaks in the Cairns region (Pratchett et al., 2014).

Similarly, the count submodel was conceived hierarchically, with a covariate (κ1) to account for potential differences between point counts and timed swim methods used within the FMP surveys: (25) λrs∼eNδrs,τδ

(26) δrs=δr0+κ1TS

(27) δr0∼Nκ0,τκ

(28) τδ,τκ=σδ−2,σκ−2

(29) σδ−2,σκ−2∼U0,1000.

Because the AMPTO CPUE data was not collected from the same segments within a reef we elected to model individual CPUE records (l) within a reef as Poisson samples, using their corresponding reef-scale averages (βr), after first calibrating using the detectability and CPUE scaling parameter (B) estimates from the mark-recapture model: (30) elogCPUErl−logϕ ¯B∼Poisλr

(31) λr∼eδr0.

In this way, the AMPTO CPUE observations were considered informative of reef-scale average CoTS densities within the Cairns sector, adding information to that present in the FMP surveys.

All models were run using the Metroplis-Hastings algorithm for 106 iterations, with a 900,000 burn in period, using the PyMC2 package (Patil, Huard & Fonnesbeck, 2010) for the Python programming language. Model convergence was assessed using Gelman–Rubin statistics from multiple model runs (Gelman & Rubin, 1992) and model fit was evaluated using Bayesian p-values (Brooks, Catchpole & Morgan, 2000), with scores lower than 0.025 and 0.975 providing substantial evidence for lack of model fit.

Results

Mark-recapture results

In total, 114 individual CoTS with disc width varying from 15 to 35 cm were observed and tagged over the eight study sites. As intuition would support for a large, slow-moving benthic invertebrate, average detectability of adult CoTS on a reef was high, at 0.82 [0.77, 0.87] (median highest posterior density (HPD) and [95% uncertainty intervals]). Given this high average detectability, the posterior ‘true’ abundance across all sites was estimated to be 116 [114, 120] individuals.

Starfish size had the greatest overall effect on detectability, with CoTS larger than 30 cm being highly detectable (P(detection) >0.8) and detection declining substantially to near zero for the smallest starfish, none of which were observed in the study (Fig. 2B). While we found little evidence for an effect of hard coral cover (Fig. 2A), some inter-site variability in detection was present, with one site (Undine 3) having markedly lower average detection than the majority of reefs (Fig. 2D). Time of day also had an effect on detectability, with CoTS being more detectable during the day than at night (Fig. 2C).

Figure 2 Factors affecting the detectability of crown-of-thorns starfish on the Great Barrier Reef.

(A) Highest posterior density (HPD) effect sizes for alternative observation team (Team), tagging effects (Tagged), nighttime surveys (Night), animal size (disc with), and the percentage of hard coral present within the survey area. (B) Estimated median relationship between animal size and detectability (solid blue line), with 95% uncertainty intervals (dashed lines) and observed detection rates for k = 6 capture-occasions (dots). (C) Posterior probabilities of detection for CoTS, given presence for surveys conducted during the day and night. (D) Posterior probabilities of detection for CoTS among survey sites.

Other measured factors were found to have limited effects on detection, with little evidence that alternative observation teams were more or less likely to detect individual CoTS (Effect size (ES) −0.13 [−0.63, 0.36] and 0.18 [−0.36, 0.72]). Unsurprisingly, given the bright white tags used, there was a modest, positive effect of tagging (ES 0.30 [−0.22, 0.80]) on CoTS detectability (Fig. 2A).

Data calibration results

Inter-calibration of the AMPTO and FMP datasets showed there were consistent biases present in both. For the AMPTO data, the relationship between CPUE and known density was shown to be hyper-stable, with a scaling exponent B of 0.33 [0.21, 0.46]. Transformed to the original scale, substantial uncertainty in this relationship remained, as only eight data points were available for estimation (Fig. 3A).

Figure 3 Data calibration of alternative CoTS survey methods.

(A) Estimated median catch-per-unit-effort (CPUE) calibration curve (solid blue line) and 95% uncertainty intervals (dotted lines) for AMPTO surveys; estimated relationship with CoTS density per hectare was DCoTS = [log(0.83) + 0.33∗log(CPUE)]∗40; (B) posterior median (dots), 50% (thick lines) and 95% (thin lines) density estimated CoTS densities for detection-corrected FMP surveys. Black diagonal is a 1:1 line in (B).

Unlike the AMPTO CPUE data, the FMP data was shown to be primarily biased by detectability, with detection-corrected estimates falling close a 1:1 line (Fig. 3B). Deviations from 1:1 were observed where substantial site-level heterogeneity had been estimated among the site-level random effects (Undine 3 and Rudder 4; Fig. 2D), suggesting some level of site-level variability remained unaccounted for in our model. However, none of our model-fit measures displayed any evidence for lack of fit given our statistical model (Bayesian p-value 0.71).

Regional density estimation

Our joint zero-inflated, Bayesian hierarchical model found that 75 (63%) of the 120 reefs surveyed likely (i.e., P(outbreak) >0.5) experienced outbreaks in 2014 (Fig. 4A), with substantially higher probabilities in the North, near Lizard Island, than among reefs further south (Fig. 4C). Average outbreak probabilities ranged from between 0.95 (i.e., more than three adult CoTS in each manta-tow segment within a reef) to 0.06 (only a 6% average chance of outbreak across manta-tow segments).

Figure 4 Data-integrated estimated CoTS outbreak densities in the Cairns sector of the Great Barrier Reef, 2014.

(A) Reef-wide average probability of a CoTS outbreak; (B) reef-wide average expected CoTS density; (C) estimated relationship between linear distance from Lizard Island and the probability of CoTS outbreak among Cairns-sector reefs.

Given the estimated outbreak probabilities, the expected density of CoTS among manta-tow segments varied substantially across reefs, with northerly (outbreaking) reefs having 62 or more CoTS per hectare (top 5% of densities), and southerly (non-outbreaking) reefs experiencing densities below 12 CoTS per hectare (bottom 5% of densities; Fig. 4B). Overall, by accounting for detection, we estimate CoTS densities among reefs within the reefs surveyed in the Cairns sector averaged 44 [41, 48] CoTS per hectare in 2014.

Discussion

The problem of imperfect species detectability has been a focus in terrestrial monitoring for decades (Kéry & Schmidt, 2008) but has received less attention in marine monitoring programs (Coggins Jr, Bacheler & Gwinn, 2014). Without recognizing the true underlying density of CoTS, management actions are much more likely to fail, due to insufficient effort allocated to find remnant populations. Furthermore, understanding and predicting CoTS outbreaks on the GBR and elsewhere has been limited by the quantity and variety of information about basic density estimates and population-level dynamics and coordination among different survey methods and effort; current surveillance for CoTS is not systematic, instead relying primarily on reports made by existing operations. Here we successfully combine the first rigorous estimates of detectability for CoTS with multiple sampling surveys undertaken within the known area of active outbreaks, to estimate the density of adult CoTS within the region. As such, this joint approach should help improve decisions concerning how CoTS outbreak and spread, and the management interventions that may be taken.

Given the tendency for CoTS to hide in crevices and under corals, we were surprised to find that the percentage of hard coral cover had little effect on the detectability of adult CoTS. However, this may be due both to the relatively narrow range of hard coral cover observed (14 to 42%) and the generally large size of CoTS (median = 35 cm); habitat complexity (which we did not measure) may have also affected detectability, however complexity levels were subjectively considered to be similar among transects. Large adult CoTS (>40 cm) are thought to be most active predominantly in daytime with smaller COTS (<20 cm) tending to be more active at night (De’ath & Moran, 1998). Thus, we were not surprised to see that they were somewhat more detectable during the day (0.85) than at night (0.75). Alternatively, increased nighttime movements could have decreased detectability if it led to some CoTS departing the survey area. Although survey divers did not report such activity, it would violate the closure assumption of our models and not allow us to partition nighttime detectability from nocturnal movement.

Detectability among adult CoTS varied most strongly with body size, from highly detectable (up to 0.98) animals with discs 40 to 50 cm wide, declining appreciably (to less than 0.5) among the smallest CoTS observed (<15 cm). This substantial decline in detectability at small body sizes that has the greatest potential to restrict understanding of CoTS population dynamics for two reasons. First, with very low expected detectability (<0.1) at the smallest body sizes it becomes exceptionally difficult to quantify annual recruitment—i.e., CoTS that survive the larval stage and settle on the reef. Secondly, the susceptibility of juvenile CoTS to fish predation is unclear (Sweatman et al., 2008), as are natural rates of post-recruitment mortality. Improved understanding of recruitment and post-recruitment mortality will be critical for the assessment of whether cull campaigns can effectively control CoTS outbreaks once they have begun.

The data-integration conducted in our study was sufficient to reveal a strong pattern of diminishing outbreaks from North to South within the Cairns sector, supporting the long held belief that northern GBR CoTS outbreaks originate near Lizard Island and progress southward (Pratchett et al., 2014), a pattern strongly dependent on subsequent waves of recruitment. It is the magnitude of annual recruitment that determines the potential spread of a given outbreak, while juvenile survival affects outbreak rate and severity, and is an as-yet untested target for tactical CoTS control.

While there are multiple benefits of increased survey accuracy at the reef level, efficiency gains in directing potential control actions can also be made by formally integrating all available monitoring information. Currently, CoTS control efforts are dedicated toward maintaining coral cover above ecologically important thresholds on a subset of 21 commercially and ecologically important reefs. Despite these efforts, there remain substantial accuracy gains to be made by adopting a formal framework for estimating CoTS densities more generally. This is clear from our illustrative example, where the integration of FMP and AMPTO surveys generated a 15% decrease in the coefficient of variation (CV) of γ0, the average overall density parameter. Although we integrated only two data sources, others—such as from the AIMS long-term monitoring program and the GBRMPA’s Eye on the Reef Program—could be easily added. Such precision increases can only help to direct limited control resources more effectively.

In this study, we have addressed two important gaps in the understanding and quantification of dynamics of CoTS on the GBR, namely in calibrating multiple data sources for bias, primarily due to method and detectability. This increased accuracy can provide immediate benefits in improving the state of knowledge and management of CoTS through the current outbreak. However, a major knowledge gap still remains in that only adult CoTS are counted and killed within current surveys. The juvenile life-history stage, from settlement (0.05 cm) to maturity (∼11 cm) (Pratchett et al., 2014), remains a crucial black box that should become a focus for future monitoring research. Without the ability to detect and kill juvenile CoTS, adult control operations will only remove the threat to corals for a single year, after which juvenile CoTS will mature and emerge from the reef substrate to feed. It is likely only through early detection of larvae (Uthicke et al., 2015), juvenile, and pre-spawn adult COTS that candidate control methods could hope to arrest initial outbreaks.

We wish to thank the FMP, AMPTO, AIMS dive teams for their professionalism and commitment to CoTS research. We would like to thank M Kayal and another anonymous reviewer for their constructive comments during peer-review. This publication incorporates data which is © Commonwealth of Australia 2014 (Great Barrier Reef Marine Park Authority). The data has been used with the permission of the Great Barrier Reef Marine Park Authority on behalf of the Commonwealth. The Great Barrier Reef Marine Park Authority has not evaluated the data as altered and incorporated within the text and therefore gives no warranty regarding its accuracy, completeness, currency or suitability for any particular purpose and no liability is accepted (including without limitation, liability in negligence) for any loss, damage or costs (including consequential damage) relating to any use of the data. AIMS thanks the Great Barrier Reef Marine Park Authority, the Queensland Government, AMPTO and all contributors to the Integrated Eye on the Reef Program for the provision of these data associated with reef health and crown-of-thorns starfish.

Additional Information and Declarations

Competing Interests

Author Contributions

Field Study Permissions

Data Availability

Steven Moon is an employee of Association of Marine Park Tourism Operators, Cairns, Australia.

M. Aaron MacNeil conceived and designed the experiments, analyzed the data, wrote the paper, prepared figures and/or tables, reviewed drafts of the paper.

Camille Mellin, Morgan S. Pratchett and Kenneth R.N. Anthony wrote the paper, reviewed drafts of the paper.

Jessica Hoey and Zara L. Cowan conceived and designed the experiments, wrote the paper, reviewed drafts of the paper.

Alistair J. Cheal, Ian Miller and Hugh Sweatman conceived and designed the experiments, performed the experiments, wrote the paper, reviewed drafts of the paper.

Sascha Taylor performed the experiments, wrote the paper, reviewed drafts of the paper.

Steven Moon performed the experiments.

Chris J Fonnesbeck analyzed the data, wrote the paper, reviewed drafts of the paper.

The following information was supplied relating to field study approvals (i.e., approving body and any reference numbers):

All fieldwork was conducted under the Australian Institute of Marine Science under Great Barrier Reef Marine Park Authority Permit G06/19994.1.

The following information was supplied regarding data availability:

A repository with our mark-recapture analysis and data is available on GitHub: https://github.com/mamacneil/CoTS_MR.

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
