# Peer review of "Joint estimation of crown of thorns (Acanthaster planci) densities on the Great Barrier Reef"

_PeerJ, doi:10.7717/peerj.2310_

## Round 0.1 · original submission · Minor Revisions

I have now heard back from two reviewers who were both positive about your report. However, both have recommended some helpful minor changes and have some questions for you, and therefore my decision is 'minor revision'. I look forward to seeing a revised version.

Reviewer 1 ·

Basic reporting

The article “Joint estimation of crown of thorns (Acanthaster plancii) densities on the Great Barrier Reef” describes basic but very significant issue: detectability of COTS in the field using traditional mark and capture methods and clarify which factors are affecting on the detectability of COTS in Great Barrier Reef. It is true that despite the fact that detectability of animal in the ocean is expected to be lower than on land, there are few studies facing this issue seriously. The conclusion that larger COTS are easier to find is not an unexpected result but still worthwhile accurately testing in the field.
The manuscript is well written and organized. I think it worth publishing after minor revisions.

In the title misspelling Acanthaster plancii ⇒ Acanthaster planci

Experimental design

Tagging COTS is sometimes difficult because they easily remove tags within a few days. Please write in more detail as to how authors tag COTS (which material was used for penetrating the body) and how long do authors expect the tag would last in the field.
It seems the survey was carried out within a month (May 2014). Are there any COTS with possibly removed the tags even with more than two tags per COTS?

Please mention the average interval (how many days) of each mark-capture survey.

Although the meanings are simple, equations are not easy to follow readers are not easy to find the meaning of each symbol.

Validity of the findings

Well analyzed and the results are very interesting and valuable as written in the manuscript.

Additional comments

I think this kind of basic work is very crucial for accurately understanding ecology.
Although the result of this work is basically applicable for GBR, I hope similar study will be done in the near future in different reefs in order to accurately estimate the number of COTS in the field.

Unfortunately, I could not understand the meaning of Fig 1d. Is it possible to explain more detail?

·

Basic reporting

The manuscript is well written and clear. There are a few points that need clarification before the manuscript is published (see comments to authors below).

Experimental design

No Comments

Validity of the findings

No Comments

Additional comments

The term “pest” as used to refer to COTS would benefit a clear definition, given that COTS are native to reefs in many regions and their populations naturally show cyclic variations. It is not clear whether the pest designation comes from the oscillating demography of this species, or to a possible increase in frequency of outbreaks related to human activities, or to the destructive predation impacts to coral populations when outbreaks occur.

Similarly, the term “true density” needs to be defined and probably be written within brackets. In particular, as small COTS individuals were neither observed (all COTS had central disks >15cm) nor searched for (highly cryptic behavior of juvenile COTS), the authors might want to refer to “known or reference adult density based on the more thorough search method used" rather than “true density”.

The authors have tested the effects of different factors on detectability of COTS. For example coral cover, time of the day, observation team, presence of tags, etc. (l. 141-143). Surprisingly habitat complexity was not measured as a covariate, although mentioned as one of the main factors influencing COTS detectability (l.136-140). I guess this is due to logistic limitations given that measuring habitat complexity would have asked for additional sampling efforts that were not part of the survey. Yet, the authors still need to mention the potential importance of habitat complexity in discussion (2nd paragraph), as it remains probably the main determinant of COTS detectability after individual size.
Overall, although coral cover and habitat complexity are often related, it seems logical that coral cover would mostly influence presence of COTS within a reef area (given targeted movement of COTS toward live coral, eg see Kayal et al. 2012 in PLoS ONE), while habitat complexity would be most influential on COTS detectability (given lower sight of observer and cryptic behavior of COTS).
Besides, weather conditions (visibility, light, swell, etc.) are also expected to affect COTS detectability, but there is no information about how far apart were the repeated observations performed. Thus it is unclear if such factors could have differed between observations and potentially affected the results of the study.


SPECIFIC COMMENTS

l. 79: the term “AUD” can be defined or spelled out.

l. 98-99: are the reef identification codes necessary? If yes, please state what they correspond to.

l.103-104: “initial surveys of the reef surface at between seven to nine meters”, precise “depth” if this is what it is about.

l.109 is missing a space after the coma.

l. 110-111: “Each transect was re-surveyed K=6 times during both day (60% of observations) and night”, how far apart in time were these repeated surveys performed? Is it within a few hours or over several days? This is important information given that probability of observing the same individuals is expected to decrease over time.

l.175: add “are” before “difficult to estimate”.

l. 205-206: “probability of a CoTS outbreak - defined as three or more CoTS or feeding scars per manta-tow (Doherty et al. 2015)”, I am surprised that feeding scars are accounted for for as much as detection of individual COTS in these observations. Shouldn’t we expect several feeding scars per COTS? Field observations often suggest so, of course depending on coral availability and habitat patchiness. For example, from Kayal et al. 2012 (PLOS ONE): “an average ratio of 8.6 ±1.7 SE feeding-scars per seastar was calculated over the process of the outbreak”.
Besides, the cited paper by Doherty et al. is missing in the Reference list of the manuscript.


l.214: remind the reader what the “B” parameter refers to.

Figure 1c: the parameters yi and zi need to be defined to make this figure self explanatory.

Figure 2b: it seems that the dots represent the probability of COTS detection over the 6 sampling events. Yet, there is not information on this in the legend. Similarly, a color code seems to be used to distinguish the 2 reefs. Additionally, the legend refers to “dotted lines” which I do not see on the plot (only the dashed lines are shown).

Figure 3c: Does the diagonal line represent the 1-1 x-y ratio? Please define in legend.

---

## Round 0.2 · accepted · Accept

Both reviewers are satisfied with the current version, and I look forward to seeing this published. Congratulations!

Reviewer 1 ·

Basic reporting

Now the manuscript is further improved and I think it is ready for publication.

Experimental design

No commnets

Validity of the findings

good

Additional comments

Great work!!

Nina Yasuda

·

Basic reporting

No Comments

Experimental design

No Comments

Validity of the findings

No Comments

Additional comments

The authors have adequately addressed all the questions raised on a previous version of this manuscript.